# The 83 symptoms of tinnitus: Content overlap of commonly used scales for tinnitus burden

Milena Engelke[1], Jorge Piano Simões[2]*, Berthold Langguth[1], Winfried Schlee[1,3], Laura Basso[1]

1 Department of Psychiatry and Psychotherapy, University of Regensburg, Regensburg, Germany,
2 Department of Psychology, Health and Technology, University of Twente, Enschede, Netherlands,
3 Institute for Information and Process Management, Eastern Switzerland University of Applied Sciences, St. Gallen, Switzerland

☯ These authors contributed equally to this work.
* j.pianosimoes@utwente.nl

## Abstract

### Background

Clinical management of tinnitus remains challenging due to unclear etiology and diverse phenotypic manifestations. To quantify its associated burden, a variety of patient-reported outcome measures (PROMs) are used. This study aims to comprehensively evaluate the content overlap of items between eight PROMs commonly used in tinnitus research.

### Methods

A two-stage, blinded multi-rater process was used to analyze the content of all 199 items from the International Tinnitus Inventory (ITI), Subjective Tinnitus Severity Scale (STSS), Tinnitus Functional Index (TFI), Tinnitus Handicap Inventory (THI), Tinnitus Handicap Questionnaire (THQ), Tinnitus Primary Function Questionnaire (TPFQ), Tinnitus Questionnaire (TQ), and Tinnitus Reaction Questionnaire (TRQ). The Jaccard Index was used to measure pairwise content overlap between scales.

### Results

The analysis revealed 83 distinct symptoms. "Concentration" was the most frequently captured symptom (in seven scales), whereas 41 symptoms (49.4%) were unique to one scale. The TQ exhibited the highest number of unique symptoms (52.5%), while the THI had the least (4%). The Jaccard Index identified very weak/weak scale overlap between the PROMs. The highest overlap was observed between TFI and THI (0.35). The TFI had the highest mean overlap (0.26), coming closest to the content measured by all other PROMs.

**Data availability statement:** Data and code are available at GitHub (https://github.com/MilenaEn/83-Symptoms-of-Tinnitus). DOI: 10.5281/zenodo.17854750.

**Funding:** The author(s) received no specific funding for this work.

**Competing interests:** The authors have declared that no competing interests exist.

**Abbreviations:** PROMs, patient-reported outcome measures; ITI, International Tinnitus Inventory; STSS, Subjective Tinnitus Severity Scale; TFI, Tinnitus Functional Index; THI, Tinnitus Handicap Inventory; THQ, Tinnitus Handicap Questionnaire; TPFQ, Tinnitus Primary Function Questionnaire; TQ, Tinnitus Questionnaire; TRQ, Tinnitus Reaction Questionnaire.

## Conclusion

The results demonstrate high heterogeneity and limited content overlap among tinnitus burden PROMs, similar to other conditions. The findings suggest that tinnitus burden is not measured as a unified construct across questionnaires, thus, researchers and clinicians should carefully consider the specific symptoms measured when selecting instruments for treatment evaluation and comparison.

## Introduction

Tinnitus is a condition characterized by the perception of sounds without a corresponding external stimulus. The condition affects 14.4% of the world population, and 2.3% of the population is debilitated by the condition [1]. The etiology of tinnitus remains incompletely understood, and although several risk factors being associated to it, the evidence for these factors is considered low. However, hearing loss, either due to presbycusis or insult to the auditory system is considered the main risk factor [2,3]. There are several neurophysiological models that explain the origin and maintenance of tinnitus [4]. A main challenge to disentangle tinnitus etiological underpinnings remains a better understanding of its heterogeneity. Given the variable clinical presentation of tinnitus, transitioning from understanding its manifestations in an individual patient to assessing its impact on his/her daily live is crucial for effective management.

Tinnitus heterogeneity has been implicated not only in etiological variability, but also in its diverse clinical manifestation. For instance, tinnitus can be accompanied by several comorbidities, all of which may require clinical care to mitigate disease burden and psychological distress [2,5–7]. As there is no established biomarker, researchers and clinicians usually resort to patient reported outcomes (PROMs) to quantify tinnitus burden. A recent systematic literature review identified 33 tinnitus-related PROMs, spanning constructs like coping, acceptance, catastrophizing and hearing problems [8]. Of these, eight measured the construct "tinnitus burden", also operationalized as tinnitus distress, tinnitus severity, negative impact of tinnitus, tinnitus handicap and tinnitus-related complaints. Next to tinnitus loudness, tinnitus distress is the primary outcome domain most often reported in clinical trials [9], and therefore a crucial component to measure psychological burden and to evaluate the effectiveness of clinical interventions. Commonly used instruments in tinnitus research and clinical practice to measure tinnitus-related burden include the Tinnitus Handicap Inventory (THI) [10], the Tinnitus Functional Index (TFI) [11], the Tinnitus Questionnaire (TQ) [12] and the Tinnitus Reaction Questionnaire (TRQ) [13]. Thus, different instruments are applied for the quantification of tinnitus-related burden, with the THI and TFI being the most widely used instruments [9].

The lack of a gold-standard instrument can be problematic as authors seldom justify why a specific questionnaire was chosen to be administered. Legacy data, historical gold standards in a subfield, the country/region where the study was conducted, language availability, copyright access, time frame of the study and type of

intervention are practical aspects which may play a role for employing certain questionnaires [9], although the justification for choosing a specific questionnaire is not reported in most cases. This gives the impression of an implicit, albeit untested assumption, that PROMs claiming to measure the same construct can be used interchangeably; not only in the tinnitus field, but across health research in general [14,15]. However, there is mounting evidence that PROMs used in psychiatric and psychosomatic research do not capture similar constructs, such as non-overlapping classification of severity [16] or different factor structures [17].

Previous research has shown that total scores of PROMs measuring tinnitus-related burden are strongly correlated (TFI, THI, THQ [Tinnitus Handicap Questionnaire] [18], TQ, TRQ) [19]. Boecking and colleagues showed that the intra-class correlation of total scores of the TQ, TFI, and THI ranges between 0.72 and 0.83 [20]. However, even if total scores of PROMs correlate, it remains unclear whether they measure the same construct. As shown by Fried, high correlation between total scores can be achieved between scales with disparate items, highlighting the relevance of also evaluating the content of individual items [15].

In this study, we investigated the extent to which item content overlaps between tinnitus-burden PROMs. To do so, we utilized the same framework proposed by Fried [15]. Thus, we posit that PROMs measuring tinnitus burden are only interchangeable to the extent that their item content exhibits overlap.

## Methods

### Data

A previous systematic review identifying questionnaires in otology was consulted to pick all relevant PROMs measuring the construct "tinnitus burden", also operationalized as tinnitus distress, tinnitus severity, negative impact of tinnitus, tinnitus handicap and tinnitus-related complaints [8], which resulted in the inclusion of the following eight questionnaires: The International Tinnitus Inventory (ITI) [21], the Subjective Tinnitus Severity Scale (STSS) [22], the Tinnitus Functional Index (TFI) [11], the Tinnitus Handicap Inventory (THI) [10], the Tinnitus Handicap Questionnaire (THQ) [18], the Tinnitus Primary Function Questionnaire (TPFQ) [23], the Tinnitus Questionnaire (TQ) [12], and the Tinnitus Reaction Questionnaire (TRQ) [13]. Throughout this manuscript, the terms PROM, questionnaire and scale are used interchangeably.

### Questionnaires

The 8-item ITI was designed as a streamlined instrument for utilization in clinical settings to shed light on the predominant complaints of tinnitus [21]. The STSS consists of 16 items and was developed to quantify tinnitus severity within a single score [22]. The 25-item TFI measures severity and negative impact of tinnitus and has been developed with a specific focus on responsiveness to treatment effects [11]. The 25-item THI is one of the most widespread tinnitus PROMs with three subscales reflecting functional, emotional and catastrophic responses of tinnitus [10]. The THQ has 27 items with three underlying factors addressing the patients' physical, emotional and social health, their hearing ability and their view on tinnitus [18]. The TPFQ is a 20-item questionnaire that queries impairment of tinnitus in the domains emotion, hearing, sleep and concentration [23]. The TQ stands as the earliest questionnaire in this series and is also the lengthiest one with 52 items measuring complaints resulting from tinnitus [12]. The TRQ consists of 26 items and was designed to assess psychological stress associated with tinnitus [13]. All questionnaires have been psychometrically validated [19–24].

### Procedure

After the relevant questionnaires had been identified, their item content was assessed in a two-stage process. We used the English version of all questionnaires in the subsequent analysis. In the first stage, the three raters [JPS], [ME] and [LB] individually labelled each of the 199 items with keywords best describing its content. The raters were blinded to each other's labels and were instructed to identify short keywords to describe each item, and to review their labels after at

least 24 hours. Individual rates took place between October and December 2023. The second stage labelling took place immediately after the first stage, as all three reviewers compared their labels and obtained consensus when necessary. Within this second step, labels were also systematically compared within and between questionnaires to ensure the utilization of uniform labels for analogue items, adhering to the conservative methodology outlined by Fried [15]. To give some examples, items were labelled uniformly if they were coded reversed (hearing clearly vs. hearing difficulty), belonged to the same category (job vs. household responsibilities) or were weighted differently (feeling ill vs. feeling terribly diseased). This consensus version was used for the final analysis. All individual rater labels, as well as the final labels, are publicly available (https://doi.org/10.5281/zenodo.17854750). For consistency with the protocol developed by Fried [15], we employ the term "symptom" to designate the label of the items.

Further, to regroup the identified symptoms, we implemented a semi-automated approach: First, a large language model was applied three times to categorize the symptoms. Second, a researcher team consisting of tinnitus experts and psychologists (ME, JPS, LB) reviewed the three versions to develop the final categories. Please note that it was the same team that rated the items in the first stage, as familiarity with the content of the questionnaires was helpful when regrouping. For the first step, we prompted Chat-GPT on the 2nd of January 2024 (version 3.5) three times to categorize them by content criteria (prompt: "Here's a list of keywords, each presented at a different row: [list of symptoms]. Please cluster them according to similar topics. All words should be assigned to only one topic"). The generated categories (first round: emotional well-being, emotional distress, attitude and perception, health and physical symptoms, communication and support; second round: sleep and sleep-related issues, auditory challenges, emotional well-being, general health issues, coping mechanisms, social and relationship impact, existential and emotional struggles, enjoyment and activities, personal development and well-being; third round: sleep-related issues, auditory disturbances, coping mechanisms, emotional distress, quality of life impact, personal perception and attitude, enjoyment and relaxation) were reviewed and synthesized by the authors and used as the basis for the final categorization, with additional consideration of categories commonly used in the literature. The established categories were then used to visualize the overlapping latent structure of the questionnaires.

## Statistical methods

Following the item rating, the Jaccard Index was used to measure pairwise content overlap between items from the questionnaires. This index is commonly used for binary data and ranges from 0 to 1, with 0 indicating no overlap, and 1 indicating complete overlap between items. It can be calculated with the following formula:

$$\frac{s}{(u_1 + u_2 + s)}$$

With $s$ representing the number of items two questionnaires are sharing, while $u_1$ and $u_2$ represent the number of items that are unique in each questionnaire. According to Fried [15], the following categorization of the Jaccard Index is adopted: very weak 0–0.19, weak 0.20–0.39, moderate 0.40–0.59, strong 0.60–0.79, and very strong 0.80–1 [25].

All the analyses were performed in R [version 4.2.2] [26] and based on the script provided by Fried [15].

## Results

The rating procedure of the content of the 199 items from eight PROMs measuring tinnitus burden resulted in 83 different symptoms (see Fig 1). Labeling of items revealed analogue items within-questionnaire for five questionnaires (STSS, THQ, TPFQ, TQ, TRQ) which resulted in a length of 174 adjusted items (see Table 1). Comparing the item labels between-questionnaire resulted in a final symptom list of 83 symptoms. Of those, 41 were idiosyncratic and appeared only in one scale. In relation to the scale length, the THI had least idiosyncratic symptoms (N = 1; 4%), while the TQ had most idiosyncratic symptoms (N = 21, 52.5%).

 

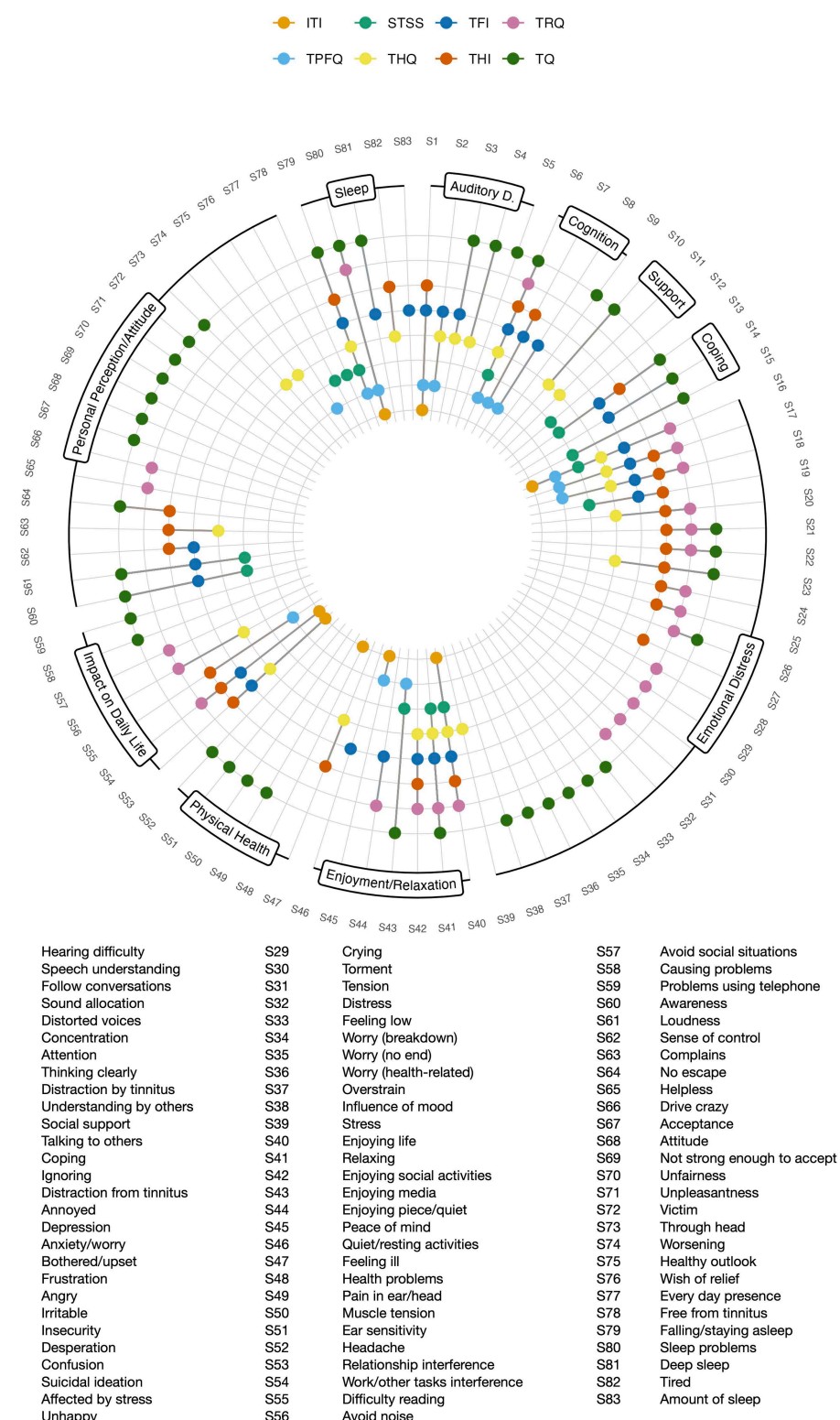

Fig 1. **Occurrence of 83 tinnitus burden symptoms across eight questionnaires.** The symptoms were regrouped and labeled according to a semi-automated approach to aid visualization. ITI (International Tinnitus Inventory), STSS (Subjective Tinnitus Severity Scale), TFI (Tinnitus Functional Index), THI (Tinnitus Handicap Inventory), THQ (Tinnitus Handicap Questionnaire), TPFQ (Tinnitus Primary Function Questionnaire), TQ (Tinnitus Questionnaire), TRQ (Tinnitus Reaction Questionnaire). Auditory D.=Auditory Disturbances.

| | | | |
|---|---|---|---|
| S1 | Hearing difficulty | S29 | Crying |
| S2 | Speech understanding | S30 | Torment |
| S3 | Follow conversations | S31 | Tension |
| S4 | Sound allocation | S32 | Distress |
| S5 | Distorted voices | S33 | Feeling low |
| S6 | Concentration | S34 | Worry (breakdown) |
| S7 | Attention | S35 | Worry (no end) |
| S8 | Thinking clearly | S36 | Worry (health-related) |
| S9 | Distraction by tinnitus | S37 | Overstrain |
| S10 | Understanding by others | S38 | Influence of mood |
| S11 | Social support | S39 | Stress |
| S12 | Talking to others | S40 | Enjoying life |
| S13 | Coping | S41 | Relaxing |
| S14 | Ignoring | S42 | Enjoying social activities |
| S15 | Distraction from tinnitus | S43 | Enjoying media |
| S16 | Annoyed | S44 | Enjoying piece/quiet |
| S17 | Depression | S45 | Peace of mind |
| S18 | Anxiety/worry | S46 | Quiet/resting activities |
| S19 | Bothered/upset | S47 | Feeling ill |
| S20 | Frustration | S48 | Health problems |
| S21 | Angry | S49 | Pain in ear/head |
| S22 | Irritable | S50 | Muscle tension |
| S23 | Insecurity | S51 | Ear sensitivity |
| S24 | Desperation | S52 | Headache |
| S25 | Confusion | S53 | Relationship interference |
| S26 | Suicidal ideation | S54 | Work/other tasks interference |
| S27 | Affected by stress | S55 | Difficulty reading |
| S28 | Unhappy | S56 | Avoid noise |

| | |
|---|---|
| S57 | Avoid social situations |
| S58 | Causing problems |
| S59 | Problems using telephone |
| S60 | Awareness |
| S61 | Loudness |
| S62 | Sense of control |
| S63 | Complains |
| S64 | No escape |
| S65 | Helpless |
| S66 | Drive crazy |
| S67 | Acceptance |
| S68 | Attitude |
| S69 | Not strong enough to accept |
| S70 | Unfairness |
| S71 | Unpleasantness |
| S72 | Victim |
| S73 | Through head |
| S74 | Worsening |
| S75 | Healthy outlook |
| S76 | Wish of relief |
| S77 | Every day presence |
| S78 | Free from tinnitus |
| S79 | Falling/staying asleep |
| S80 | Sleep problems |
| S81 | Deep sleep |
| S82 | Tired |
| S83 | Amount of sleep |

**Table 1.  Distribution of symptoms per scale.**

|  | ITI | STSS | TFI | THI | THQ | TPFQ | TQ | TRQ | Sum/Mean (SD) |
|---|---|---|---|---|---|---|---|---|---|
| **Original scale length (No.)** | 8 | 16 | 25 | 25 | 27 | 20 | 52 | 26 | 199 |
| **Adjusted scale length (No.)** | 8 | 14 | 25 | 25 | 23 | 14 | 40 | 25 | 174 |
| **Idiosyncratic symptoms (No.)** | 1 | 3 | 2 | 1 | 4 | 1 | 21 | 8 | 41 |
| **Idiosyncratic symptoms (%)** | 12.5 | 21.4 | 8 | 4 | 17.4 | 7 | 52.5 | 32 | 19.4 (16.2) |

Original scale length (No.): Number of items. Adjusted scale length (No.): Number of symptoms after uniform labelling of analogue items. Idiosyncratic symptoms (No.): Number of symptoms that appear in no other scale. Idiosyncratic symptoms (%): Percentage of symptoms that appear in no other scale (Idiosyncratic symptoms/ Adjusted scale length). ITI (International Tinnitus Inventory), STSS (Subjective Tinnitus Severity Scale), TFI (Tinnitus Functional Index), THI (Tinnitus Handicap Inventory), THQ (Tinnitus Handicap Questionnaire), TPFQ (Tinnitus Primary Function Questionnaire), TQ (Tinnitus Questionnaire), TRQ (Tinnitus Reaction Questionnaire).

From a symptom-level perspective, a symptom was measured on average in two scales (Median = 2, SD = 1.5). 41 symptoms (49.4%) occurred only in one scale, while one symptom (1.2%) appeared in seven scales (Concentration) and three symptoms (3.6%) were identified in six scales (Annoyed, Falling/staying asleep and Enjoying life). Anxiety/worry, Depression and Relaxing appeared in five scales. There was no symptom that was featured across all eight scales (see Table 2).

According to the Jaccard Index, the scale overlap between the relevant PROMs ranges from very weak to weak (0.02–0.35; see Table 3), with the highest overlap among TFI and THI (0.35) and the lowest overlap between ITI and TQ (0.02). The TFI had the highest mean overlap with other questionnaires (0.26), followed by the THI (0.23). The TQ showed the lowest mean overlap with other questionnaires (0.11). On average, the investigated PROMs had a Jaccard Index of 0.18 (SD = 0.05), which corresponds to a very weak mean overlap of the scales.

The following 10 higher-level categories were used to group the symptoms: *Emotional Distress, Auditory Disturbances, Sleep, Cognitive Disturbances, Personal Perception/Attitude, Coping, Health/Physical Symptoms, Enjoyment/Relaxation, Impact on Daily Life, Communication/Support*. The allocation of symptoms and questionnaires to those categories are shown in Fig 1. Overall, the categories *Emotional Distress* (24 symptoms, captured by eight scales) and *Personal Perception/Attitude* (19 symptoms, captured by seven scales) were covered the most, while the categories *Coping* (three symptoms, captured by four scales) and *Communication/Support* (three symptoms, captured by three scales) were covered the least; see Table 4. The categories *Emotional Distress, Enjoyment/Relaxation,* and *Sleep* were the only ones captured by all eight questionnaires.

## Discussion

In this study, we investigated the content overlap of eight questionnaires measuring tinnitus burden. We identified 83 symptoms from a total of 199 items, indicating high symptom heterogeneity and little content overlap. These findings are aligned

**Table 2.  Number of symptoms that appeared across number of scales.**

| Number of symptoms | % of symptoms | Number of scales |
|---|---|---|
| 41 | 49.4 | 1 |
| 18 | 21.7 | 2 |
| 11 | 13.3 | 3 |
| 6 | 7.2 | 4 |
| 3 | 3.6 | 5 |
| 3 | 3.6 | 6 |
| 1 | 1.2 | 7 |
| 0 | 0 | 8 |

Reading of the table (first row): 41 of the 83 symptoms (49.4%) appeared across one scale.

**Table 3. Scale overlap.**

| | ITI | STSS | TFI | THI | THQ | TPFQ | TQ | TRQ |
|---|---|---|---|---|---|---|---|---|
| **ITI** | | | | | | | | |
| **STSS** | 0.10 | | | | | | | |
| **TFI** | 0.18 | 0.30 | | | | | | |
| **THI** | 0.14 | 0.15 | 0.35 | | | | | |
| **THQ** | 0.11 | 0.16 | 0.30 | 0.33 | | | | |
| **TPFQ** | 0.22 | 0.17 | 0.30 | 0.22 | 0.19 | | | |
| **TQ** | 0.02 | 0.17 | 0.16 | 0.12 | 0.12 | 0.08 | | |
| **TRQ** | 0.14 | 0.11 | 0.22 | 0.28 | 0.23 | 0.15 | 0.10 | |
| **Mean overlap** | 0.13 | 0.17 | 0.26 | 0.23 | 0.21 | 0.19 | 0.11 | 0.18 |

*Note.* Scale overlap is quantified by the Jaccard Index, with 0 indicating no overlap, and 1 indicating complete overlap between items. ITI (International Tinnitus Inventory), STSS (Subjective Tinnitus Severity Scale), TFI (Tinnitus Functional Index), THI (Tinnitus Handicap Inventory), THQ (Tinnitus Handicap Questionnaire), TPFQ (Tinnitus Primary Function Questionnaire), TQ (Tinnitus Questionnaire), TRQ (Tinnitus Reaction Questionnaire).

**Table 4. Coverage of higher-level categories by symptoms and scales.**

| Category | Number of symptoms | Number of scales |
|---|---|---|
| *Emotional Distress* | 24 | 8 |
| *Perception/Attitude* | 19 | 7 |
| *Enjoyment/Relaxation* | 7 | 8 |
| *Impact on Daily Life* | 7 | 7 |
| *Health/Physical Symptoms* | 6 | 4 |
| *Sleep* | 5 | 8 |
| *Auditory Disturbances* | 5 | 6 |
| *Cognitive Disturbances* | 4 | 7 |
| *Coping* | 3 | 4 |
| *Communication/Support* | 3 | 3 |

with the ones previously found in depression [15], sleep disorder [27], mental health [14], trauma [28], mental pain [29], neurological soft signs [30], obsessive compulsive disorder [31], mania [32], anxiety [33], and romantic-relationship [34].

The highest yet still weak overlap between questionnaires based on the Jaccard Index was observed among TFI and THI (0.35) and the lowest overlap between ITI and TQ (0.02). The TFI had the highest mean overlap with the other questionnaires (0.26), the TQ showed the lowest mean overlap with the other questionnaires (0.11). The highest mean overlap suggests that the TFI comes closest to the content measured by all other PROMs. However, we would like to stretch that this is not a direct measure of content validity as we cannot exclude the possibility that important aspects of subjective suffering are not depicted in any of the questionnaires.

The most featured symptoms across the eight questionnaires were: *concentration* (present in 7 PROMs), *enjoying life*, *falling/staying asleep*, *annoyed* (present in 6 PROMs), *anxiety/worry*, *depression*, *relaxing* (present in 5 PROMs). Notably, five of these symptoms were identified as core outcome domains of interest in a previous Delphi study ("concentration", "quality of sleep", "tinnitus intrusiveness", "negative thoughts/beliefs", "mood") [35]. Almost half of the 83 symptoms (49.4%), including, e.g., acceptance, torment, and avoiding social situations, were only featured in one PROM. The TQ contained the most idiosyncratic symptoms (21 symptoms, i.e., 52.5% of the questionnaire), which can be partially explained by its length (52 items compared to the average of 21 items among the other 7 PROMs included in the analysis). This implies that the TQ assesses many symptoms that are not featured by other PROMs.

Notably, we found surprising few symptoms related to somatic complaints. Of the investigated PROMs, only the TQ featured a few somatic symptoms (Pain in ear/head, Headache, Muscle tension). Its relation with tinnitus burden is well established in the literature, with a previous study reporting 42% of patients with somatic symptom disorder also suffering from tinnitus [36]. For instance, dizziness and hyperacusis are commonly occurring somatic comorbidities which, despite their clinical relevance [37,38], are not part of any of the PROMs investigated. Likewise, the relationship between tinnitus and pain has been previously established in empirical [39] and theoretical [40,41] works. Moreover, patients with hyperacusis and chronic pain were found to have higher TFI scores [42], highlighting the impact of somatic symptoms on tinnitus burden even without being directly measured. However, it remains an open question whether somatic symptoms directly characterize tinnitus burden and should thus be incorporated into a PROM assessing tinnitus burden, or whether these instruments should focus strictly on tinnitus-specific burden and not assess any related (somatic) comorbidities.

Our finding, indicating that tinnitus burden is not measured as a unitary construct but often encompasses various idiosyncratic symptoms, carries another significant implication. Previous research investigated tinnitus heterogeneity in terms of the diverse acoustic presentation, the broad spectrum of comorbidities or unique sociodemographic risk factors [43,44]. Additional to the interindividual variability in tinnitus related symptoms, the use of different outcome measures might lead to different study results. This may serve as a contributing factor to the empirical findings of low consistencies between responder rates among different PROMs [20,45], acknowledging that there may be additional explanations. A practical consequence of this finding is that tinnitus-burden PROMs should not be assumed to be interchangeable.

Based on our analysis, the overlap among the PROMs under investigation ranges from weak to very weak. Consequently, comparing results across trials that employ different outcome measures should either be avoided or approached with great caution. Currently, the THI and the TFI are the most frequently utilized PROMs in clinical tinnitus trials [9]. Standardizing the use of these PROMs in future studies could enhance the comparability of results. Additionally, when evaluating interventions targeting specific symptom domains, it may be advantageous to incorporate supplementary questionnaires that are designed to assess those particular domains more comprehensively.

The interpretation of this analysis should take into account the subjective nature of the rating process, acknowledging that different raters may have arrived at alternative conclusions. Nevertheless, adhering to the methodology outlined by Fried [15], we maintained a conservative approach, suggesting that the number of symptoms collected by the analyzed PROMs is likely underestimated. Simultaneously, a different set of PROMs analyzed would have led to another result. The selection of PROMs was based on their inclusion in a comprehensive review, and we posit that the chosen PROMs effectively represent the questionnaires commonly employed for measuring tinnitus burden. In addition, the items were not checked for completeness, i.e., whether they comprehensively measure the construct tinnitus burden, but only according to their frequency and overlap in PROMs. It's essential to also recognize that questionnaire data provide only snapshots in time, failing to capture the dynamic and fluctuating nature of tinnitus symptoms along with their associated burden [46]. Moreover, identified symptoms were clustered by topic in a bottom-up manner (see Fig. 1). Future research could compare these data-driven categories with conceptual categories from PROMs.

This work is further limited by the lack of comparison with empirically reported symptoms of tinnitus burden. A recent study labeled and categorized answers from 678 patients to the question *Why is tinnitus a problem?* [47]. They identified 18 problem domains of which "Reduced quality of life" was most frequently represented. While this symptom is at least not directly covered by the PROMs analyzed here, seven items related to the impact on daily life were identified across seven PROMs. Following that, "Fear," "Constant Awareness", "Annoyance," and "Inability to Concentrate" accounted for the majority of reports. Notably, while concentration is collected by almost every tinnitus burden PROM, awareness is only incorporated in three out of eight questionnaires. Further, a patient survey found loudness reduction to be most relevant from a patients' perspective, which was only featured across three PROMs (TFI, TQ, STSS) [48]. A systematic comparison between symptoms sampled by PROMs and those reported by patients could establish external validity offering valuable

insights for stakeholders in the selection of PROMs. We believe that the patients' perspective is crucial for the development of relevant outcome measurements.

## Conclusion

As demonstrated, we found considerable symptom heterogeneity and limited content overlap across tinnitus burden PROMs. The highest overlap, albeit weak, was found between the TFI and the THI. This finding has important practical implications: tinnitus-burden PROMs should not be assumed to be interchangeable. Consequently, we strongly encourage researchers and clinicians to make informed, domain-based and patient-centered decisions when selecting PROMs.

## Author contributions

**Conceptualization:** Milena Engelke, Jorge Piano Simões, Laura Basso.

**Formal analysis:** Milena Engelke, Jorge Piano Simões, Laura Basso.

**Supervision:** Berthold Langguth, Winfried Schlee.

**Writing – original draft:** Milena Engelke, Jorge Piano Simões, Laura Basso.

**Writing – review & editing:** Berthold Langguth, Winfried Schlee.

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
