## [Decision Letter · Decision Letter 0]

27 Nov 2025

Dear Dr. Simões,

Thank you for submitting your manuscript to PLOS ONE. After careful consideration, we feel that it has merit but does not fully meet PLOS ONE’s publication criteria as it currently stands. Therefore, we invite you to submit a revised version of the manuscript that addresses the points raised during the review process.

We look forward to receiving your revised manuscript.

Kind regards,

Paul H Delano, Ph.D.

Academic Editor

PLOS ONE

Journal Requirements:

“There was no specific funding for this project.”

Reviewers' comments:

Reviewer's Responses to Questions

**Comments to the Author**

1. Is the manuscript technically sound, and do the data support the conclusions?

Reviewer #1: Yes

Reviewer #2: Yes

2. Has the statistical analysis been performed appropriately and rigorously?

Reviewer #1: Yes

Reviewer #2: Yes

3. Have the authors made all data underlying the findings in their manuscript fully available?

Reviewer #1: Yes

Reviewer #2: Yes

4. Is the manuscript presented in an intelligible fashion and written in standard English?

Reviewer #1: Yes

Reviewer #2: Yes

Reviewer #1: The manuscript by Simões, J et al. presents a well-structured and clearly written study that constitutes a meaningful contribution to the field. The authors effectively address the important research question of content and overlap among tinnitus questionnaires assessing patient burden. This analysis offers a valuable step toward a consensus on PROMs selection and interpretation, thereby facilitating more direct and reliable comparisons between future studies, which is especially vital for clinical trial outcomes.

Specific comments

It is important to note that all the selected questionnaires and subsequent analyses are conducted in English. This should be explicitly mentioned in the manuscript to ensure clarity for readers regarding the scope and applicability of the findings.

Ln115 and 131: JPS or JS? I assume this refers to the same author (Simões, J.). The reference should be made consistent throughout the manuscript.

Ln 131: I suggest clarifying that this is the same team as in the first stage, in order to ensure transparency for the readers. Additionally, it would be helpful to explain the reason for this selection.

Ln155: I strongly recommend citing R software as mentioned by them https://intro2r.com/citing-r.html.

Conclusion: It may be important to include information about the overlap (even if limited) between the TFI and the THI, as these are the most widely used PROMs worldwide, both in research and clinical settings. The following key statement from the discussion could also be highlighted in this section as a take-away message: “A practical consequence of this finding is that tinnitus-burden PROMs should not be assumed to be interchangeable.”.

Overall, the comments are minor and should be straightforward to address.

Reviewer #2: This is a rarely case of a manuscript, simple, elegant in design and perfectly delivered. My only suggestion is to improve the aesthetics of the tables, but this is only a fomating suggestion. But I would go forward to publication.

**Do you want your identity to be public for this peer review?** For information about this choice, including consent withdrawal, please see our Privacy Policy

Reviewer #1: No

Reviewer #2: **Yes:** Hayo A. Breinbauer

---

## [Author Response · Author response to Decision Letter 1]

8 Dec 2025

Response to Reviewers

Reviewer #1:

The manuscript by Simões, J et al. presents a well-structured and clearly written study that constitutes a meaningful contribution to the field. The authors effectively address the important research question of content and overlap among tinnitus questionnaires assessing patient burden. This analysis offers a valuable step toward a consensus on PROMs selection and interpretation, thereby facilitating more direct and reliable comparisons between future studies, which is especially vital for clinical trial outcomes.

Response: We would like to sincerely thank the reviewer for the time spent on our manuscript!

Specific comments

It is important to note that all the selected questionnaires and subsequent analyses are conducted in English. This should be explicitly mentioned in the manuscript to ensure clarity for readers regarding the scope and applicability of the findings.

Response: We totally agree with this valuable remark and included a corresponding note within the methods (line 113).

Ln115 and 131: JPS or JS? I assume this refers to the same author (Simões, J.). The reference should be made consistent throughout the manuscript.

Response: Thank you for reading carefully. We made the reference consistent.

Ln 131: I suggest clarifying that this is the same team as in the first stage, in order to ensure transparency for the readers. Additionally, it would be helpful to explain the reason for this selection.

Response: We hope we ensured more transparency by adding this information in line 130, thank you for the suggestion!

Ln155: I strongly recommend citing R software as mentioned by them https://intro2r.com/citing-r.html.

Response: Thank you for mentioning this! Very important remark to value their work! We cited accordingly.

Conclusion: It may be important to include information about the overlap (even if limited) between the TFI and the THI, as these are the most widely used PROMs worldwide, both in research and clinical settings. The following key statement from the discussion could also be highlighted in this section as a take-away message: “A practical consequence of this finding is that tinnitus-burden PROMs should not be assumed to be interchangeable.”.

Response: Thank you for pointing this out! We have added both points to the conclusion (lines 287–289).

Overall, the comments are minor and should be straightforward to address.

Reviewer #2:

This is a rarely case of a manuscript, simple, elegant in design and perfectly delivered. My only suggestion is to improve the aesthetics of the tables, but this is only a fomating suggestion. But I would go forward to publication.

Response: Thank you very much for taking the time to read our manucript and for your kind words! While we agree with your criticism of the tables' appearance, we would like to point out that they were formatted according to the journal's specifications.

---

## [Editor Report · Decision Letter 1]

22 Dec 2025

The 83 symptoms of tinnitus: Content overlap of commonly used scales for tinnitus burden

PONE-D-25-35632R1

Dear Dr. Simões,

We’re pleased to inform you that your manuscript has been judged scientifically suitable for publication and will be formally accepted for publication once it meets all outstanding technical requirements.

Kind regards,

Paul H Delano, Ph.D.

Academic Editor

PLOS One
---

## [Editor Report · Acceptance letter]

PONE-D-25-35632R1

PLOS One

Dear Dr. Simões,

I'm pleased to inform you that your manuscript has been deemed suitable for publication in PLOS One. Congratulations! Your manuscript is now being handed over to our production team.

Kind regards,

on behalf of

Dr. Paul H Delano

Academic Editor

PLOS One